# Wound Restorative Power of *Halimeda macroloba*/ Mesenchymal Stem Cells in Immunocompromised Rats via Downregulating Inflammatory/Immune Cross Talk

**DOI:** 10.3390/md21060336

**Published:** 2023-05-30

**Authors:** Eman Maher Zahran, Reham H. Mohyeldin, Fatma Mohamed Abd El-Mordy, Sherif A. Maher, Omar Y. Tammam, Entesar Ali Saber, Faisal H. Altemani, Naseh A. Algehainy, Mohammad A. Alanazi, Mohammed M. Jalal, Mahmoud A. Elrehany, Usama Ramadan Abdelmohsen

**Affiliations:** 1Department of Pharmacognosy, Faculty of Pharmacy, Deraya University, Minia 61111, Egypt; eman.maher@deraya.edu.eg; 2Department of Pharmacology, Faculty of Pharmacy, Deraya University, Minia 61111, Egypt; reham.hassan@deraya.edu.eg; 3Department of Pharmacognosy and Medicinal Plants, Faculty of Pharmacy (Girls), Al-Azhar University, Cairo 11754, Egypt; 4Department of Biochemistry, Faculty of Pharmacy, New Valley University, New Valley 72713, Egypt; 5Department of Medical Science, Histology and Cell Biology, Faculty of Pharmacy, Deraya University, New Minia City 61111, Egypt; 6Department of Medical Laboratory Technology, Faculty of Applied Medical Sciences, University of Tabuk, Tabuk 71491, Saudi Arabia; 7Department of Biochemistry, Faculty of Pharmacy, Deraya University, Minia 61111, Egypt

**Keywords:** *Halimeda macroloba*, stem cells, wound healing, inflammatory mediators, immunocompromise

## Abstract

Impaired skin wound healing is still a major challenge, especially with immunocompromised patients who express delayed healing and are susceptible to infections. Injection of rat-derived bone marrow mesenchymal stem cells (BMMSCs) via the tail vein accelerates cutaneous wound healing via their paracrine activity. The present work aimed to investigate the combined wound-healing potential of BMMSCs and *Halimeda macroloba* algae extract in immunocompromised rats. High-resolution liquid chromatography-mass spectrometry (HR-LC-MS) investigation of the extract revealed the presence of variant phytochemicals, mostly phenolics, and terpenoids, known for their angiogenic, collagen-stimulating, anti-inflammatory, and antioxidant properties. The BMMSCs were isolated and characterized for CD markers, where they showed a positive expression of CD90 by 98.21% and CD105 by 97.1%. Twelve days after inducing immunocompromise (40 mg/kg hydrocortisone daily), a circular excision was created in the dorsal skin of rats and the treatments were continued for 16 days. The studied groups were sampled on days 4, 8, 12, and 16 after wounding. The gross/histopathological results revealed that the wound closure (99%), thickness, density of new epidermis and dermis, and skin elasticity in the healed wounds were considerably higher in the BMMSCs/*Halimeda* group than the control group (*p* < 0.05). RT-PCR gene expression analysis revealed that the BMMSCs/*Halimeda* extract combination had perfectly attenuated oxidative stress, proinflammatory cytokines, and *NF-_K_B* activation at day 16 of wounding. The combination holds promise for regenerative medicine, representing a revolutionary step in the wound healing of immunocompromised patients, with still a need for safety assessments and further clinical trials.

## 1. Introduction

The process by which damaged tissue is replaced by freshly formed cells in a living organism is known as wound healing. The wound healing process is subdivided into inflammation, proliferation, extracellular matrix formation, and finally remodeling [1]. The immune system regulates tissue healing and regeneration while circulating leucocytes are drawn to the injury site by signals sent by the damaged cells, causing natural killer cells, T lymphocytes, dendritic cells, neutrophils, and monocytes to become active in the bloodstream. Immediately after injury, innate immune cells release variant inflammatory cytokines, such as (interleukin -1-beta (*IL-1β*), interleukin 6 (*IL-6*), and interleukin 8 (*IL-8*)), tumor necrosis factor-α (*TNF-α*), and chemokines, which start the inflammatory response’s acute phase. According to estimates from the WHO, about 5 million people may die every year as a result of improper wound healing [2]. Impaired wound healing can be attributed to chronic exposure to excessively high levels of glucocorticoids and usually results in immunosuppression [1]. Additionally, steroid-induced oxidative damage may also impair wound healing because it results in excessive amounts of reactive oxygen species (ROS), impaired ROS detoxification, and altered inflammatory profiles, which are the main causes of non-healing chronic wounds [2].

Mesenchymal stem/stromal cells (BMMSCs) are pluripotent T-cells with differentiation capacity and immunomodulatory characteristics [3]. They play a central role in tissue repair, in addition to having antifibrotic, antitumorigenic, antiapoptotic, proangiogenic, neuroprotective, anti-inflammatory, antibacterial, and chemo-attractive properties [3]. The therapeutic potential of BMMSCs in the areas of regenerative medicine, inflammatory disorders, and increasingly cancer therapy is made appealing by this set of characteristics. Initially, BMMSCs were used for illness treatment as well as tissue regeneration and repair in graft-vs-host disease (GVHD) and autoimmune diseases. Additionally, the clinical potential of BMMSCs has been expanded to include the treatment of cancer, diabetes, cirrhosis, multiple sclerosis, myocardial infarction, and stroke. Several tissue types, including adipose tissue, fetal tissue, dental pulp, umbilical cord, and placental tissue, have been used to extract BMMSCs, but the majority of preclinical research has used bone marrow-derived stem cells (BMMSCs) [4,5]. Recently, it has been demonstrated that BMMSCs can modulate immune responses, taking part in both innate and adaptive immunity through interactions with T-regulatory cells and monocytes [6]. They have the extraordinary capacity to differentiate into many types of body cells during early life and growth. They can be further categorized into pluripotent (embryonic) stem cells and nonembryonic (adult stem cells), which serve as the body’s system for repairs [6]. Adult stem cells (ASCs) differentiate to produce the specialized cell types of a tissue type or organ, whereas embryonic stem cells can grow into every form of adult body cell. Since they are viewed as a source for replacing cells lost during wound repair, adult stem cells (ASCs) are recognized as important participants in tissue regeneration [7]. BMMSCs are classified into CD34^+^ and CD34^−^: CD34^+^ cells produce blood cells, while CD34^−^ cells can differentiate into many cell types [8]. In general, BMMSCs are low in frequency in the bone marrow; and human bone marrow-derived BMMSCs can be cloned and expanded in vitro more than a million-fold, producing large numbers of BMMSCs for cell therapy. Consequently, BMMSCs have already been included in safety studies and for early clinical testing for the treatment of a variety of diseases.

Since ancient times, the marine environment has long been utilized for the richness of secondary metabolites, especially seaweed, which have been used as a diet component in several areas of the world [9]. Within different seaweed species, there are varied antioxidant properties, which result from the presence of compounds such as carotenoids, polyphenolics, and terpenoids [10]. Among many seaweeds, *Halimeda macroloba* has been verified among phytopharmaceuticals, owing to its antioxidant, cytotoxic, antimalarial, hepatoprotective, and antimalarial properties [11]. The majority of the secondary metabolites found in *Halimeda,* which have not yet been properly investigated, are phenolics. This green seaweed also is reported to contain halogenated compounds, terpenes, and sterols, which are reported to have radical scavenging and anti-inflammatory potencies. Due to their astringent, antibacterial, free radical scavenging, and lipid peroxidation inhibitory properties, phenols aid in wound healing by preventing cell damage and increasing the survivability of collagen fibrils [12]. Traditional medicine is still widely practiced throughout the world’s less-developed regions, particularly for the treatment of wounds [13]. Collectively, stem cell therapy for wound healing has developed into a highly promising and sophisticated scientific study area in recent years, and *Halimeda* has never been investigated for wound healing. Consequently, we aimed in the current study to explore the metabolomic profiling of the green seaweed *Halimeda macroloba* using LC-HRESIMS and assess its potential for wound healing in immunocompromised rats, side by side with the injection of BMMSCs, derived from rats, via histopathological as well as PCR analyses.

## 2. Results

### 2.1. Metabolomic Profiling of Halimeda macroloba Algae

Metabolic profiling of *Halimeda macroloba* algae extract using LC-HRESIMS was carried out to tentatively identify its secondary metabolites. Eighteen secondary metabolites were identified from *Halimeda macroloba* algae (Appendix A and Figure 1) that belong to a variety of chemical classes. Identification of the annotated compounds was performed via comparison with the data reported in the literature.

The annotation revealed the presence of vanillic acid 4-O-sulfate **(1)**, which was previously detected in *Sargassum seaweed* species [14]; catechin **(2)** and gallocatechin **(10)**, which were previously reported in *Caulerpa* spp. [14]; dihydro-caffeic acid 4-*O*-glucuronide **(3)** [15]; protocatechuic acid 4-*O* glucoside **(4)**, already detected in *Centroceras* spp. [14]. Additionally, 3-O-(6’-sulfo-α-D-quinovopyranosyl)-glycerol **(5)** has previously been reported in the green alga *Ulva pertusa* [16]. *Halimeda macroloba* has previously yielded 4-*O*-(40-(dimethylamino)-40-iodobutan-50-yl-10,20,30-triol)-N methylbutanamide **(6)**, di-(2-ethylhexyl) phthalate **(17)**, and 24-isopropyl cholesterol **(19)** [11]. Other pyridoacridines that have been found in the marine tunicate *Cystodytes violatinctus* [17] include cycloshermilamine D **(7)**, galloyl glucose **(8)** [15], and the diterpenoids Helmidatetraacetate **(11)** and Halimedatrial **(12)**, which were previously found in the genus *Halimeda* (Chlorophyta) [18]. Finally, in addition to 2-methoxy-6Z-octadecenoic acid **(13)** [18] and chlorophyll B **(14)**, three steroids (clionasterol **(15)**; cholesta-5,22-dien-3beta-ol (22-dehydrocholesterol) **(16)** and cholestane-3beta,5a-diol-6-one **(18)**) [19,20,21] were previously reported in *Halimeda opuntia* [22].

### 2.2. Biological Study

#### 2.2.1. Gross Evaluation and Estimation of Wound Closure Rate

The results of gross observation at 0, 4, 8, 12, and 16 days showed that the wound closure rate was greatly enhanced in the treated groups, mostly observed in group 6, where the wounds had nearly disappeared at day 16, as shown in Figure 2A. Figure 2 also illustrates the semiquantitative wound closure rate of the six groups (*n* = 6), where (B) represents the percentage of wound closure calculated by ImageJ with time after injury (days 4, 8, 12, and 16). To help the wound tissue close, the centripetal flow of the edges of a full-thickness wound can be used to identify wound closure. On the other hand, Figure 2C highlights group 6 as attaining the best aspect shape ratio, which refers to the differences in the shape and direction of wound contraction that were observed between groups. Group 6 expressed the combined effects of using both the extract and the BMMSCs in the form of the highest wound closure rate and the lowest aspect shape ratio, where the skin appeared nearly normal.

#### 2.2.2. BMMSC Immunophenotyping

After bone marrow isolation, a flow cytometric analysis of the BMMSC surface markers was performed to confirm that these cells were BMMSCs, not hematopoietic stem cells. The results showed a positive expression of mesenchymal markers, CD90 by 98.21% and CD105 by 97.1%, and a negative one of CD34 and CD45 markers (Figure 3). These markers were used to verify that the isolated BMMSCs were not hematopoietic stem cells, following the recommendations of the International Society for Cellular Therapy’s Mesenchymal and Tissue Stem Cell Committee [23].

#### 2.2.3. Histopathology Results

The histopathological results of the examined sections were coincident with those obtained from the gross evaluation as follows.

Group I (negative control rats): The wound extends the full thickness of the skin. The surface of the wound was filled with blood clots, and the base was filled with granulation tissue formed of several layers of disorganized compact collagen, connective tissue cells in an acidophilic matrix, and overlying heavy inflammatory cellular infiltration attached to the dilated blood capillaries and extravasated RBCs (Figure 4(1)). At the wound edges, normal epidermis, dermal connective tissue with well-formed collagen bundles, normal hair follicles, and sebaceous glands were seen.

Group II (immunosuppressed rats): The wound surface was filled with hemostatic clots, while the wound bed was much larger than it seemed on the surface. The wound bed was filled with sloughed granulation tissue, with edema, cellular debris, extravasated RBCs, and inflammatory cell infiltration, mainly eosinophils and neutrophils. Disorganized compactly arranged collagen bundles and abnormal epidermal cell growth and/or rolled wound edges were often observed (Figure 4(2)).

Group III (immunosuppressed rats + standard "market" preparation): The wound was covered with dense-crust tissue blocking the wound and a single layer of epidermal cells had formed over the damaged area trying to cover the wound. The dermis showed congested blood capillaries, blood clots blocking the damaged blood vessels, and neovascularization in the upper part of the granulation tissue. Migration of inflammatory cells, mainly macrophages, was observed, and discontinuous collagen fibers had appeared, filling the bottom of the wound (Figure 4(3)).

Group IV (immunosuppressed rats + *Halimeda* extract): The wound appeared with closer edges, but was blocked with crust tissue. A deformed epidermis made of one to three cell layers without keratin covered the area of granulation tissue that filled the defect. The papillary dermis, which contained recently created collagen bundles and hair follicles, seemed thin irregular, and disorganized. Thin scar tissue extended into the dermis (Figure 4(4)).

Group V (immunosuppressed rats + mesenchymal stem cell-treated): Apparent wound contraction and incomplete re-epithelialization were observed (Figure 4(5)).

The primary cells that make the new matrix required to repair the structure and shape of injured tissue are fibroblasts. In the granulation tissue at the site of the wound, there was a distinct presentation of active elongated, spindle-shaped fibroblasts with basophilic cytoplasm and phagocytic cells with acidophilic cytoplasm. Reticular collagen bundles were arranged in a manner like that of the adjacent dermis.

Group VI (immunosuppressed rats + mesenchymal stem cells + formulated extract ointment): The skin tissue looked to be mostly normal, with the typical stratified squamous keratinized epithelium covered by thin scar tissue. The dermal matrix had numerous blood capillaries, hair follicles, and was not infiltrated by inflammatory cells. The collagen bundles appeared as thin interlacing bundles in the papillary dermis and as coarse wavy bundles arranged in different directions in the reticular dermis (Figure 4(6)).

The best results were obtained by combining the positive effects of both the extract and the BMMSCs, where the skin tissue appeared nearly normal.

#### 2.2.4. Gene Expression Results

Effect of *Halimeda macroloba extract* and mesenchymal stem cells on expression of *Cox-1*, *Cox-2*, *IL-1β*, *TNF-α*, *INF-ϒ*, *NF-KB*, *TGF-β*, and *IL-10*.

Figure 5 depicts the mRNA expression of some markers following excisional wound therapy with *Halimeda macroloba* extract, stem cells, Mebo^®^, or stem cells with *Halimeda macroloba* extract. *Cox-1*, and *Cox-2* relative mRNA expression in skin tissue was substantially lower in stem cells and *Halimeda macroloba* extract-treated wounds at 8 or 16 days compared to the untreated group (*p* < 0.05). The relative expression of stem cells and *Halimeda macroloba* extract-treated wounds, on the other hand, showed a significant decrease in marker expression compared to the Mebo^®^-treated group (Figure 5A). In Figure 5B, the mRNA expression of *TNF-α* and *IL-1β* was illustrated. Analysis of mRNA expression of full-thickness wound samples on day 8 post-injury revealed that the activity of the inflammatory markers *TNF-α* and *IL-1β* was significantly downregulated in wounds treated with stem cells and *Halimeda macroloba* extract or Mebo^®^ compared to the untreated wounds. However, wounded rats treated with stem cells and *Halimeda macroloba* extract displayed significantly much more reduction in the inflammatory markers (*TNF-α*, and *IL-1β*) when compared to the Mebo^®^-treated group. Moreover, stem cells and *Halimeda macroloba* extract treatment or Mebo^®^ treatment for 16 days showed a significantly dramatic decrease in *TNF-α* and *IL-1β* mRNA expression when compared to the untreated group at (*p* < 0.05). Again, the expression of *TNF-α* and *IL-1β* in stem cells and *Halimeda macroloba* extract-treated wounds were markedly lower than in the Mebo^®^-treated group.

The relative gene expression of *INF-ϒ* and *NF-_K_B* is illustrated in Figure 5C. Analysis of the relative expression of INF-ϒ and NF-_K_B in full-thickness wound samples on day 8 post-injury showed significantly downregulated levels in wounds treated with stem cells and *Halimeda macroloba* extract or Mebo^®^ compared to the untreated wounds. However, compared to rats treated with Mebo^®^, injured rats treated with stem cells and *Halimeda macroloba* extract showed a much greater downregulation in relative gene expression. Moreover, stem cells and *Halimeda macroloba* extract treatment or Mebo^®^ treatment for 16 days showed significantly greater decrease in relative gene expression when compared to untreated wounds at (*p* < 0.05). Again, the relative expression of *INF-ϒ* and *NF-KB* in stem cells and *Halimeda macroloba* extract-treated wounds was markedly higher than Mebo^®^-treated wounds. Figure 5D depicts the mRNA expression of anti-inflammatory cytokines *TGF-β*, and *IL-10* following excisional wound therapy with stem cells and *Halimeda macroloba* extract and Mebo^®^. *TGF-β* and *IL-10* relative mRNA expression in skin tissue was substantially higher in stem cells and *Halimeda macroloba* extract-treated wounds at 8 or 16 days compared to the untreated group (*p* < 0.05). On the other hand, compared to the Mebo^®^-treated group, the relative expression of stem cells and wounds treated with *Halimeda macroloba* extract indicated a considerable increase in marker expression.

## 3. Discussion

The metabolic identification results of *Halimeda* extract revealed the presence of a wide variety of secondary metabolites, with an abundance of phenolic compounds and terpenoids, which was encouraging. Phenolics are reported to attain anti-inflammatory, antioxidant, immunomodulatory, and wound-healing properties [24]. In addition, terpenoids have long been used in animal models for their analgesic effects, as well as antioxidant, wound healing, and antimicrobial effects [25]. Recently, an increasing number of research articles have been focusing on the utilization of natural products as possible wound healers. In fact, this was encouraging to our research, since their low price, easy availability, and few adverse effects are the principal advantages of these botanical remedies [26].

The gross evaluation outputs showed that the topical application of *Halimeda* extract on the excision wounds in the immunocompromised rats, formerly injected with BMMSCs, resulted in a significant (*p* < 0.05) reduction in wound area correlated to the untreated wounds. Nearly similar results were obtained in the groups treated individually with the extract or the BBMMSCs, but their combination resulted in a great enhancement in the wound closure rate, which is a sign of angiogenesis, keratinocyte differentiation, fibroblast proliferation, and re-epithelialization. This is coincident with the fact that wound healing is expressed in three phases: an inflammatory process caused by the secretion of proinflammatory mediators and immune system suppression, a proliferative phase via the proliferation of fibroblasts, the growth of collagen, and the development of new blood vessels, as well as a remodeling phase that includes regeneration and damaged tissue repair [1].

The complex interactions between cells and a variety of growth factors are necessary for wound-healing processes. Throughout all stages of wound healing, *TGF-β* and *IL-10* play a key role. *TGF-β* and *IL-10* trigger the recruitment and activation of inflammatory cells (such as neutrophils and macrophages) during the hemostasis/inflammation phase, while during the proliferative phase, they induce a variety of cellular responses, such as re-epithelialization, angiogenesis, granulation tissue development, and the deposition of extracellular matrix [27]. Additionally, they encourage the proliferation and differentiation of fibroblasts into myofibroblasts, which take part in wound healing during the remodeling phase. According to one study, non-healing wounds frequently worsen a failure of *TGF-β*/*IL-10* warning, while another study stated that *TGF-β* and *IL-10* exert an inhibitory effect on the expression of collagenases [28]. This is clearly consistent with the results of our work, which highlighted that BMMSCs/*Halimeda* extract enhanced *TGF-β* and *IL-10* expression compared to the untreated wound tissue.

On the other hand, proinflammatory cytokines (*IL-1β* and *TNF-α*), which have been identified as dynamic inducers of metalloproteinase (MMP) synthesis, must be expressed appropriately to recruit neutrophils and reduce contaminations from the wound site [1]. During the healing process, in order to facilitate wound repair, the MMP breaks down and eliminates damaged extracellular matrices (ECMs), yet elongation of the inflammatory phase causes difficulty with healing, since these cytokines cause tissue damage and make the wounds chronic. The top growth factor *TNF-α* is released by macrophages and combines with *IL-1β* to inhibit collagen synthesis and fibroblast proliferation. In turn, *NF-κB* and *INF-ϒ* are then boosted by *TNF-α*, which encourages gene expression of a variety of proinflammatory cytokines, including *TNF-α* itself and MMP [1]. Accordingly, suppressing inflammatory cytokines (*TNF-α,* and *IL-1β*) can inhibit chronic inflammation and enhance wound repair (Figure 6).

Mesenchymal stem cells (BMMSCs) are multipotent, self-renewing stem cells that can be found in practically all post-natal organs. Despite the fact that the processes by which BMMSCs reduce skin lesions have been hotly debated for years, two theories may now account for the therapeutic actions of BMMSCs: bioactive soluble factors (growth factors, cytokines, and specific proteins) or BMMSC differentiation into dermal and epidermal cells [3].

Some studies revealed that systemically administered BMMSCs frequently migrate to damaged sites and elicit leukocyte responses, neutrophils, macrophages, and lymphocytes. It has been also demonstrated that the MSC secretome can alter the response of macrophages, polarizing them to be transformed from proinflammatory to anti-inflammatory during skin wound healing, being a crucial step [5]. They are also said to suppress the *NF-KB* signal transduction pathway, lower apoptosis, and downregulate the expression of proinflammatory cytokines. Moreover, the use of BMMSCs promotes fibroblast migration and survival, as well as ECM deposition, proliferation, and migration [3].

Finally, BMMSCs exert potent immunomodulatory and angiomodulatory characteristics. In response to inflammatory cytokines after injury, BMMSCs induce the generation and expansion of immunosuppressive M2 macrophages and Tregs, which reduces both the negative immune response and ongoing inflammation. BMMSCs also regulate the phenotype and function of immune cells that participate in tissue repair [29]. By producing a large number of immunomodulatory molecules (such as *TGF-β*, *NO*, *IL-10*, *IL-6*, *IL-1*, *PGE2*, *TNF-α*, and VEGF) in order to improve the repair of damaged tissue, BMMSCs control immunoresponse and vasculogenesis.

## 4. Materials and Methods

### 4.1. Metabolomic Analysis of Halimeda macroloba Extract

#### 4.1.1. Algae Sample Collection and Identification

*Halimeda macroloba* specimens were gathered in May 2021 from the shorelines at Savage City, Egypt, on the Red Sea coast. To remove any contaminants, sand, or salts, the obtained samples were first washed with seawater, then with tap water, and finally with distilled water. The samples were stored in sterile plastic bottles, kept cold, and delivered to the lab in an ice box. A voucher specimen (2021-BuPD 82) was banked at the Department of Pharmacognosy, Faculty of Pharmacy, Beni-Suef University, Egypt, where the specimens were graciously recognized using common taxonomic keys.

#### 4.1.2. Extraction of *Halimeda macroloba*

About 0.25 kg of *Halimeda macroloba* samples were collected, air-dried in the shade for a month, dried, and then ground into a fine powder using an OC-60B/60B grinder (60–120 mesh, Henan, Mainland China). The powder was extracted with five liters of 70% ethanol macerated at room temperature for three days each and concentrated under vacuum at 45 °C using a rotary evaporator (Buchi Rotavapor R-300, Cole-Parmer, Vernon Hills, IL, USA) to provide 50 g of crude extract [11].

#### 4.1.3. Metabolomic Analysis

High-resolution liquid chromatography–mass spectrometry (HR-LC-MS) tentative metabolic identification was performed using a Synapt G2 HDMS quadrupole hybrid mass spectrometer (Waters, Milford, CT, USA), while the compounds were identified using the Dictionary of Natural Products (DNP) database as reported [30].

### 4.2. Biological Study

#### 4.2.1. Incision Wound Model

A total of 24 healthy male albino rats (average of 150 g each) were acclimatized to the experiment, which was performed in compliance with the *Guidelines for the Care and Use of Laboratory Animals* of the National Institutes of Health, under the approval of 19 July 2022 of the ethics committee at Deraya University [31]. Rats were randomly distributed into 6 groups of 4 animals each, where the incision wound model was performed as follows.

Group 1: Negative control rats received a topical application of the vehicle of the plant extract twice daily for 16 days after wounding and intramuscular (i.m.) injection of pyrogen-free sterile water, the vehicle of hydrocortisone (HC).

Group 2: Immunosuppressed rats (40 mg/kg HC daily, for 16 days prior to wounding) received a topical application of the vehicle of the plant extract twice daily for 12 days after wound creation.

Group 3: Immunosuppressed rats (40 mg/kg HC daily, for 16 days prior to wounding) and topical application of MEBO^®^ ointment (5% *w*/*w*) as a standard preparation (market) twice daily for 16 days after wound creation.

Group 4: Immunosuppressed rats (40 mg/kg HC daily, for 16 days prior to wounding) and topical application of formulated extract (extract suspended in carboxymethyl cellulose) twice daily for 16 days after wound creation.

Group 5: Immunosuppressed rats (40 mg/kg HC daily, for 16 days prior to wounding) and BMMSCs (0.5 × 10^6^ cells) were intravenously (i.v.) administered via the lateral tail vein of the rat following 24 h of wound creation.

Group 6: Immunosuppressed rats (40 mg/kg HC daily, for 16 days prior to wounding) and BMMSCs (0.5 × 10^6^ cells) were i.v. administered via the lateral tail vein of the rat following 24 h of wound creation plus topical application of the formulated extract ointment twice daily for 16 days after wound creation.

MEBO^®^ is the name of a market pharmaceutical product. It is an oil-based ointment containing sesame oil, beta-sitosterol, berberine, and other small quantities of plant ingredients. The beta-sitosterol, the main ingredient of MEBO^®^ has shown anti-inflammatory effects, while berberine has demonstrated antimicrobial effects.

Many studies substantiated the claim that it promotes epithelial repair, inhibits bacterial growth, attains analgesic effects, leads to reduction of water evaporation from burn wound surface, provides the optimum physiological environment for healing, and results in improved scar formation. As such, we used MEBO^®^ as a market positive control group [32].

#### 4.2.2. Excisional Wound Creation

An intraperitoneal injection of a ketamine–xylazine mixture was used to anesthetize the animals. (90 mg/kg body wt. ketamine and 10 mg/kg body wt. xylazine) before wound creation. The rat’s dorsal surface was shaved using an electric fur clipper, and 70% ethanol was sprayed over it to clean the underlying skin. A full-thickness excision wound (approximately 1 cm × 1 cm) was created using a sterile surgical blade and scissors under aseptic conditions, followed by topical application of a pain reliever (buprenorphine, 0.5 mg/kg body wt.). To mitigate the pain threshold, all efforts were made to minimize suffering. The wound was left uncovered during the whole period of the experiment. At 16 days post-wounding, the wound tissue was carefully collected under light anesthesia using ketamine (80 mg kg^−1^, i.p.) to conduct biochemical, molecular, and histopathological analysis [33].

#### 4.2.3. Induction of Immunocompromised Condition

This was accomplished by administering hydrocortisone (HC) (40 mg/kg body weight, intramuscularly) daily for one week before wounding, which was continued until the end of the trial to preserve the animals in an immunosuppressed state, as previously reported [34]. HC was dissolved in pyrogen-free sterile water.

#### 4.2.4. Collection of Tissue Samples, Calculation of Percentage Wound Closure Rate

On the 16th day, full-thickness skin biopsies of entire ulcers from all groups were collected under anesthesia. Tissue samples were sectioned into three parts for gene expression analyses and histological examination after storing in formalin.

The progression of the wound area was observed using a camera (Fuji, S20 Pro, Yama, Japan) every three days until the wound had entirely healed. The wound area was evaluated using ImageJ 1.49v software (National Institutes of Health, Bethesda, MD, USA), and the wound closure rate was calculated as percentage change in the original wound area using the following formula:$$wound\;closure\;\left(\%\right)=\frac{\left(Area\;of\;wound\;on\;day\;0-Area\;of\;wound\;on\;day\;nth\right)}{\left(Area\;of\;wound\;on\;day\;0\right)}\times 100$$
where *n* represents the examination day, i. e., 4, 8, 12, and 16.

### 4.3. Isolation, Culture, Characterization, and Injection of Mesenchymal Stem Cells

#### 4.3.1. Bone Marrow Isolation of BMMSCs

To isolate the rat bone marrow, rats were killed by cervical dislocation, the animal skeleton was rinsed with 70% ethanol, an incision was made around the perimeter of the hind limbs where they attach to the trunk, and the skin removed by pulling toward the foot. Then, the hind limbs from the trunk were carefully dissected by cutting along the spinal cord, and limbs were stored on ice in Dulbecco’s modified Eagle’s medium (DMEM) supplemented with 1× penicillin–streptomycin.

Cutting through the knee joint was performed, the muscle and connective tissue were removed from both the tibia and the femur, then the tissue pulled toward the ends of the bone. After cleaning, the bones were stored in DMEM supplemented with 1× penicillin–streptomycin on ice. Using a proper sterile technique, bone marrow was harvested, where the ends of the tibia and femur were cut just below the end of the marrow cavity. A 27-gauge needle attached to a 10 mL syringe containing complete media was inserted into the spongy bone exposed by removal of the growth plate. In a 10 mL iced tube, the marrow flush was plugged out of the cut end of the bone with 1 mL of complete media and collected. The cell suspension was filtered via a 70 mm mesh and the yield and viability of cells were determined by Trypan blue exclusion and counted on a hemocytometer [35]. The stem cell media and tools were purchased from Thermo Fisher Scientific Inc. Waltham, MA, USA).

#### 4.3.2. Culture of BMMSCs

Bone marrow cells were cultured in 1 mL of complete medium at a density of 25 × 10^6^ cells, and the plates were incubated at 37 °C with 5% CO_2_ in a humidified incubator. The non-adherent formed cells were eliminated by replacing the medium with a fresh one, a step that was repeated every 8 h for up to 72 h. Using phase-contrast microscopy on day 3, the initial adherent spindle-shaped cells appeared as individual cells, while after 2 weeks, the cells were washed with phosphate-buffered saline and incubated in 0.5 mL of 0.25% trypsin for 2 min at room temperature. The trypsin was neutralized by adding 1.5 mL of complete medium, and all lifted cells were cultured in a 25 cm^2^ flask [36].

#### 4.3.3. Trypan Blue Exclusion Test of Cell Viability

The cell suspension was centrifuged for 5 min at 100× *g*, discarding the supernatant. The cell pellet was resuspended in 1 mL PBS or serum-free complete medium. A 1:1 ratio of 0.4% trypan blue and 1 part cell suspension (1 × 10^6^ cell/ml) were mixed and incubated for 3 min at room temperature. A drop of the trypan blue–cell mixture was applied to a hemacytometer, where the unstained (viable) and stained (nonviable) cells were counted separately [37]. The percentage of the viable cells was calculated as follows:$$\mathrm{viable}\;\mathrm{cells}\;(\%)=\frac{\mathrm{Total}\;\mathrm{no}.\;\mathrm{of}\;\mathrm{viable}\;\mathrm{cells}\;\mathrm{per}\;\mathrm{ml}\;\mathrm{aliquot}}{\mathrm{Total}\;\mathrm{no}.\mathrm{of}\;\mathrm{cells}\;\mathrm{per}\;\mathrm{ml}\;\mathrm{aliquot}\;}\times \;100$$

#### 4.3.4. Characterization of BM-MSCs by Flow Cytometric Phenotyping Analysis

Flow cytometry was performed using the following mouse anti-human antibodies: anti-CD34 FITC, anti-CD45 FITC, anti-CD90 FITC, and anti-CD105 FITC. Cells were incubated with the respective antibody for 15 minutes at 4 °C, then washed with PBS and analyzed with a FACSCalibur with Cell Quest software (Becton Dickinson) (Appendix A).

#### 4.3.5. Injection of MSC

Animals were housed individually during the experiment and the BMMSCs (1 × 10^6^ cells in 60 μL of vehicle) were injected in the rat tail vein slowly. A patch (BM, Dokki, Cairo, Egypt) was immediately placed over the wounds [38].

### 4.4. Histopathological Study

Dorsal skin samples were gathered and settled in buffered formalin before being handled through a graded list of alcohol and xylene and then inserted in paraffin blocks. To assess the density of collagen fibers, tissue strips were cut 5–6 µm thick and colored with hematoxylin–eosin and a particular Masson’s trichrome. Prepared slides were studied and photographed using the Leica Application Suite (Leica Microsystems, a light microscope, Wetzlar, Germany) [1].

### 4.5. Gene Expression Analysis

Total ribonucleic acid (RNA) was separated from skin tissue by applying the TRIzol reagent (Invitrogen, Waltham, MA, USA) as per the manufacturer’s instructions. The quantity of RNA extracted was assessed spectrophotometrically using a NanoDrop 1000 (Thermo Scientific, Carlsbad, CA, USA). Complementary deoxyribonucleic acid (cDNA) was reverse-transcribed using 1 µg total RNA by a high-capacity reverse-transcription kit (Thermo Scientific, Carlsbad, CA, USA) with oligo-dT primers.

Transcript levels were established utilizing real-time polymerase chain reaction (PCR) with the sequence-specific primers listed in Appendix A. Amplification was achieved in a StepOne real-time PCR thermal cycler (Thermo Fisher, Waltham, MA, USA) as per the manufacturer’s instructions controlling the SYBR Green PCR Master Mix (Thermo Scientific, Carlsbad, CA, USA). After normalization to GAPDH as a housekeeping gene, the gene expression levels were assessed by applying the comparative CT method [1].

### 4.6. Statistical Analysis

To determine whether there was a significant difference between the groups, two-way ANOVA was performed. After one-way analysis of variance (ANOVA), the Bonferroni post hoc test was used. The data are presented as the mean ± SD. To clarify observed differences in the shape and direction of wound contraction between groups, the wound aspect ratio was calculated (length:width).

## 5. Conclusions

In immunocompromised individuals, bone marrow mesenchymal stem cells have considerable potential for promoting wound healing, tissue regeneration, and angiogenesis. Clinical approaches have shown tremendous promise for BMMSC therapy in the management of wounds. The combination of both BMMSCs and *Halimeda macroloba* extract and obtaining benefit from their immunomodulatory and repairing properties are a great challenge for modern regenerative medications, especially in complicated cases. They could amplify the downregulating effect on *Cox-1*, *Cox-2*, *IL-1β*, *TNF-α*, *INF-ϒ*, and *NF-KB*, as well as the upregulating effect on *TGF-β*, and *IL-10***.** However, more research is required to identify and classify the top stem cell sources and the most efficient strategies for delivering cells to treat wounds.

## Figures and Tables

**Figure 1 marinedrugs-21-00336-f001:**
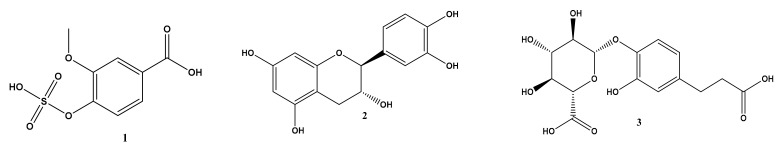
Structures of identified metabolites from the LC-HRESIMS analysis of *Halimeda macroloba* algae.

**Figure 2 marinedrugs-21-00336-f002:**
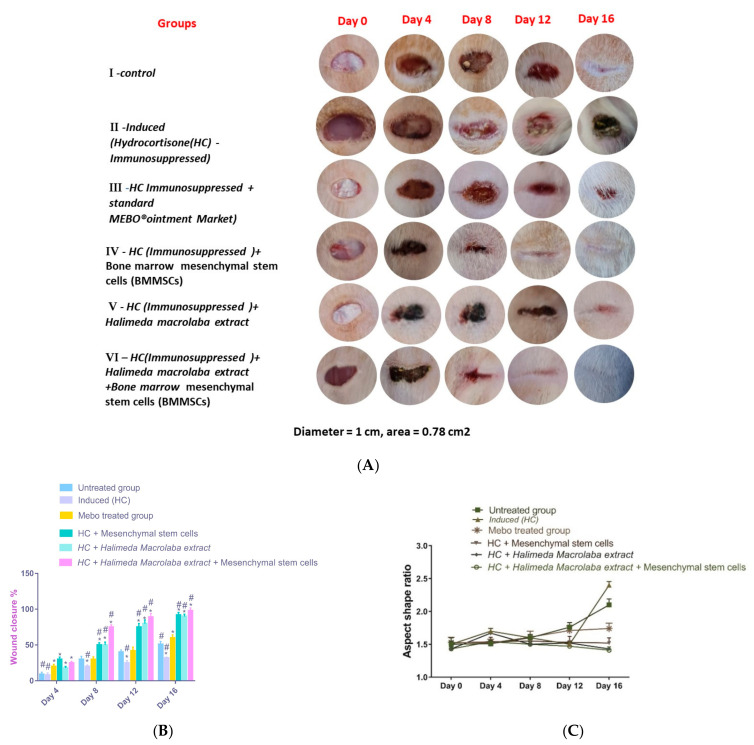
(**A**) Photographs show progression during healing of the full-thickness dermal wounds at different time intervals in excision wound models in rats. Bars represent mean ± SD. After one-way analysis of variance (ANOVA), the Bonferroni post hoc test was used to determine significant differences between the groups, where * *p* < 0.05 compared with those of the untreated group on the respective day and # *p* < 0.05 compared with those of the Mebo^®^ group on the corresponding day. (**B**) Wound closure ratio; (**C**) aspect shape ratio.

**Figure 3 marinedrugs-21-00336-f003:**
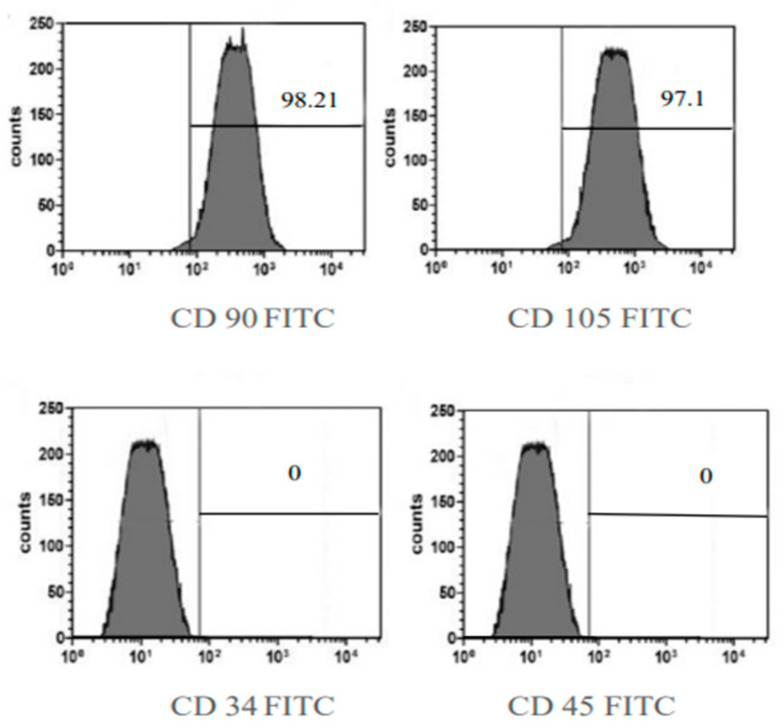
Flow cytometric analysis of cultured immunophenotype of BMMSC showing negative expression of CD34 and CD 45 and showed positive expression of CD90 (98.21%), CD105 (97.1%) antibody staining.

**Figure 4 marinedrugs-21-00336-f004:**
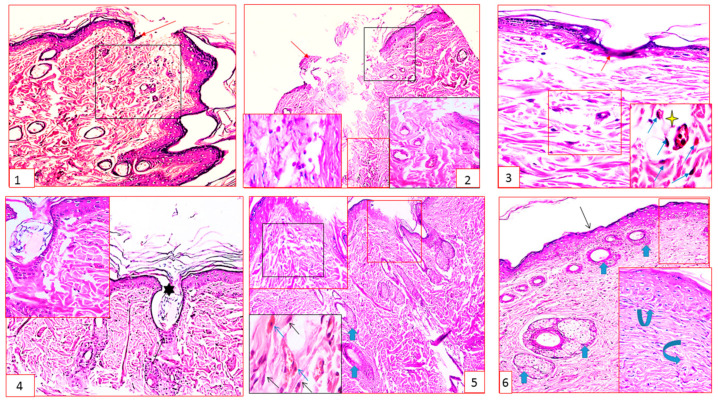
(**1**) (**Group I**), showing wound skin and the wound edges (arrow), the underlying granulation tissue in an acidophilic matrix overlying heavy inflammatory cellular infiltration (square); (**2**) (**Group II**), showing the wound bed much larger than it seems on the surface (square). The wound bed was filled with sloughed granulation tissue with edema (4-point star), cellular debris (black inset), and inflammatory cell infiltration, mainly eosinophils and neutrophils (red inset). Notice the rolled wound edge (arrow). (**3**) (**Group III**), showing crust tissue blocking the wound (star), a single layer of epidermal cells covering the wound (red arrow), granulation tissue (inset) showing congested blood capillaries, inflammatory cellular infiltration mainly macrophages (blue arrows); (**4**) (**Group IV**), showing the wound with closer edges but blocked with crust tissue (star). Granulation tissue filling the defect covered by distorted epidermis formed of 1–3 cell layers (arrow). (**5**) (**Group V**), showing apparent wound contraction (red rectangle). The underlying granulation tissue (blue square) shows marked elongated and spindle-shaped fibroblasts with basophilic cytoplasm and macrophages with acidophilic cytoplasm (black and blue arrows). (**6**) (**Group VI**), demonstrating skin tissue that seemed generally normal and had a typical epithelium. The dermal matrix has many hair follicles (thick arrows). The inset shows the papillary dermis (curved up arrows) and the reticular dermis (curved right arrows). (H&E stain × 200 and 400).

**Figure 5 marinedrugs-21-00336-f005:**
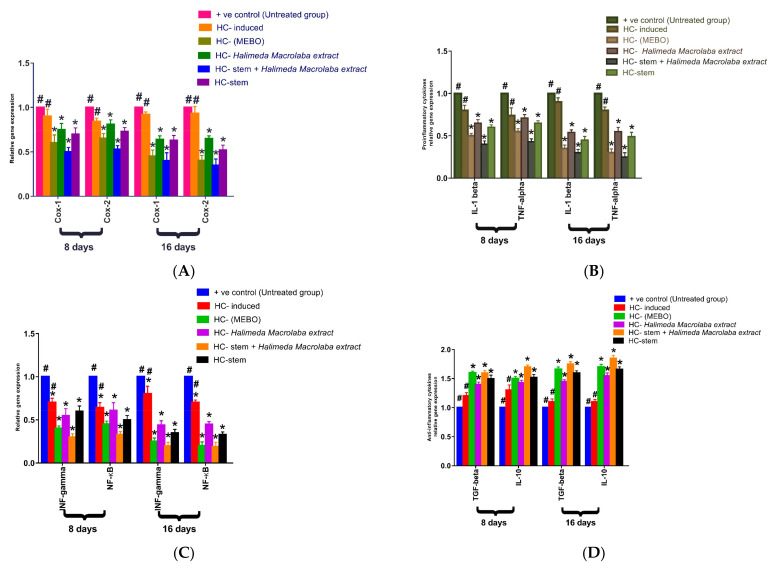
(**A**) Gene expression of *COX-I and COX-II*; (**B**) gene expression of *IL-1β* and *TNF-α*; (**C**) gene expression of *INF-γ* and *NF-_K_B*; (**D**) gene expression of *TGF-β* and *IL-10* via quantitative RT-PCR. Data reflect fold change in relation to the expression in the normal control group after being normalized to glyceraldehyde 3-phosphate dehydrogenase (*GAPDH*). Bars represent mean ± SD. After one-way analysis of variance (ANOVA), the Bonferroni post hoc test was used to determine significant differences between the groups: * *p* < 0.05 compared with those of the untreated group on the respective day and # *p* < 0.05 compared with those of the Mebo^®^ group on the corresponding day.

**Figure 6 marinedrugs-21-00336-f006:**
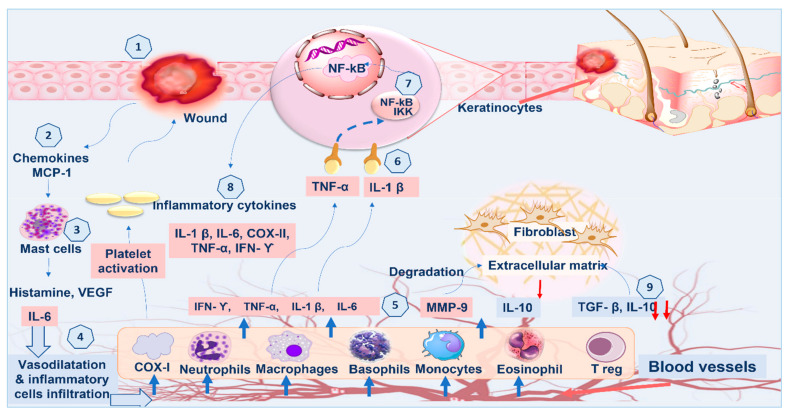
Predicted cross talk between inflammatory and immune cells in wounded/immunocompromised rats.

## Data Availability

Data are contained within the article or Appendix A.

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
