# Peer review of "Wound Restorative Power of Halimeda macroloba/ Mesenchymal Stem Cells in Immunocompromised Rats via Downregulating Inflammatory/Immune Cross Talk"

_marinedrugs, 2023, doi:10.3390/md21060336_

Round 1
Reviewer 1 Report
Review
The manuscript, “Wound restorative power of Halimeda macrolaba/mesenchymal stem cells in immunocompromised rats via Downregulating Inflammatory/Immune cross talk” by Zahran et al. aimed to investigate the combined wound-healing potential of BMMSCs and Halimeda macrolaba algae extract in immunocompromised rats. The combination of both BMMSCs and Halimeda macrolaba extract and getting the benefit of their immunomodulatory and repairing properties hold a great challenge for modern regenerative medications, especially in complicated cases.
Comments and Suggestions for Authors:
1- In the abstract, it should be said that the stem cells were taken from rats.
2- Please correct the keywords according to MeSH
3- Please write the full form of the word "HR-LC-MS" in line 28 and also "ROS" in line 56.
4- In line 103, "Stem cell therapy", the letter "S" should not be capitalized.
5- Why is the graphical abstract given in the introduction and under the title of Figure 1?
Also, figure 1 needs some changes.
An arrow has been drawn from the image of algae to stem cells. While stem cells have nothing to do with algae.
In one part of the image, a photo of the algae and the steps related to extract preparation and finally the resulting extract should be given.
In the other part, the image of the steps related to cell culture and the resulting stem cells will be given.
Finally, the products of the previous two processes should be used together on the rat.
6- In the section "4.2. The biological study", the ethical code of this study should be given. This study was conducted on rats and having an ethical code is mandatory for this study.
7- The section "4.2.1. The incision wound model", is incomplete. In this section, the conditions of keeping rats should be mentioned. Also, how to create the wound model should be fully described. The conditions for creating anesthesia in rats, the wound-causing tool, the type of wound and its dimensions, as well as the location of the wound, should be explained.
8- In line 389, write the complete form of "i.m injection".
9- In lines 401 and 404, correct the number "0.5x106cell" and write the power of 6 correctly.
10- In line 401, "intravenously (i.v)" is written, so in line 404, its full form does not need to be repeated.
11- How did mice become immunosuppressed? Please explain.
12- Lines 411-407 are related to statistical analysis and should not be included in this section. In the material and method part of this article, no part is dedicated to statistical analysis. At the end of the material and method, add a section titled "Statistical Analysis" and put these lines in that section.
13- In the method section, explain the extract that is finally used on the rats. In line 405, explain more about "formulated extract ointment" and specify the ingredients of this ointment. For example, is the pure extract used on the rat, or is the extract combined with another substance and state the concentration of this extract?
14- In the method section, it is said that BMMSCs are taken from rats. Please state this issue in the introduction section so that the reader knows the origin of BMMSCs from the beginning of the article and does not confuse it with human types.
15- The article has writing errors and needs to be revised in this regard. For example, in lines 436 and 437, "10-ml" should be written as "10 ml". In line 439, "70-mm" should be written as "70 mm". Capital letters should not be used in the middle of a sentence. For example, "Flush" in line 438, "Filtered" in line 439, "Cultured" in line 442, "Incubated" in line 443, and similarly in other parts of the article, these items should be corrected and the capital letter changed to small.
16- What do you mean by "complete medium" in line 442? Please state the name of the company and the country of this item. In the material and method section, there are many materials and tools used without the name of the company and country, which should be corrected. For example, in line 452, company and country "PBS" should be said.
17- BMMSCs culture is mentioned in the method section, but there is no image of the cell culture in the results section. Add images related to cell culture in the results section.
18- In line 422 of the method, which is prepared in the explanation of the timing of the ????? ???????, the 3rd, 7th, 10th, and 14th days are given as an example, but in line 135 of the results section, the results at other times (0, 4, 8, 12 and 16 days) has been checked. Please correct the evaluation times of ????? ??????? in the method section.
19- Figure 3 is a graphical evaluation of Figure 2. Merge these two figures together.
In Figure 2, the columns of groups are not well written. What does (-ve) mean in group 1? Group 2 should be written as "induced hydrocortisone (HC)-immunosuppressed". In group 3, "MEBO®" ointment is written, while in the method section of line 395, "povidone iodine ointment" is written, and the name of "MEBO®" ointment is not given. Please write the name of the ointment and the company and country of manufacture in the method section. In groups 4 and 6, only Mesenchymal stem cell is written. Please write the type of stem cells. Also, add "immunosuppressed" in groups 3, 4, 5, and 6.
Also in Figure 2, the images of wounds do not have scale bars, so the scale bars of the images should be given.
The description of Figure 2 in lines 148-153 should be corrected and it should be specified that rat bone marrow stem cells were used.
The description of each group for Figure 3 should be corrected as in Figure 2. Figure 3 has no explanation. Please add suitable explanations for these diagrams below the figure.
20- Figure 5 shows the histopathology images on which day? In the method section in line 413, it was said that the samples were taken for histopathological evaluation on days 7 and 14, but in the results section, only the results of one day were shown. Please add the pictures of both days 7 and 14.
Lines 215-231, which are the descriptions of Figure 5, should be given by mentioning the number of each image (1 to 6) in the continuation of line 214 and the description of the figure.
21- The discussion part has started similar to the introduction, while this part should start with the most important and main result of this study and expand, and previous studies related to this result should be reviewed and then other results of this study should be discussed in order of importance.
Author Response
Respected reviewer,
Thanks for your efforts reviewing our manuscript entitled “ Wound restorative power of Halimeda macrolaba/mesenchymal stem cells in immunocompromised rats via Downregulating Inflammatory/Immune cross talk”, ID: marinedrugs-2333700.
The responses are displayed point by point, and highlighted in yellow in the main manuscript. We hope that we could have addressed all the required comments.
Reviewer 1
The manuscript, “Wound restorative power of Halimeda macrolaba/mesenchymal stem cells in immunocompromised rats via Downregulating Inflammatory/Immune cross talk” by Zahran et al. aimed to investigate the combined wound-healing potential of BMMSCs and Halimeda macrolaba algae extract in immunocompromised rats. The combination of both BMMSCs and Halimeda macrolaba extract and getting the benefit of their immunomodulatory and repairing properties hold a great challenge for modern regenerative medications, especially in complicated cases.
Comments and Suggestions for Authors:
- In the abstract, it should be said that the stem cells were taken from rats. Thanks for your comment Sir., the required word has been added
- Please correct the keywords according to MeSH Thanks for your comment Sir., the required key words have been modified
- Please write the full form of the word "HR-LC-MS" in line 28 and also "ROS" in line 56. Thanks for your comment Sir., the required words have been modified
- Note: line 56 is now 59
- 4- In line 103, "Stem cell therapy", the letter "S" should not be capitalized. Thanks for your comment Sir., the required has been done (line 106)
- Why is the graphical abstract given in the introduction and under the title of Figure 1?
It is corrected and given under the caption: Graphical abstract
- Also, figure 1 needs some changes. An arrow has been drawn from the image of algae to stem cells. While stem cells have nothing to do with algae. In one part of the image, a photo of the algae and the steps related to extract preparation and finally the resulting extract should be given. Thanks for your comment Sir., the required changes have been made
- 6- In the section "4.2. The biological study", the ethical code of this study should be given. This study was conducted on rats and having an ethical code is mandatory for this study. Thanks for your comment Sir., it has been added and highlighted.
7- The section "4.2.1. The incision wound model", is incomplete. In this section, the conditions of keeping rats should be mentioned. Also, how to create the wound model should be fully described. The conditions for creating anesthesia in rats, the wound-causing tool, the type of wound and its dimensions, as well as the location of the wound, should be explained.
Thanks for the comment, the excisional wound creation has been made as follows:
An intraperitoneal injection of a ketamine-xylazine mixture was used to anesthetize the animals. (90 mg/kg body wt. ketamine and 10 mg/kg body wt. xylazine) before wound creation. The rat's dorsal surface was shaved using an electric fur clipper, and 70% ethanol was sprayed over it to clean the underlying skin. A full-thickness excision wound (88 mm) was created using a sterile surgical blade and scissors under aseptic conditions, followed by topical application of a pain reliever (Buprenorphine, 0.5 mg/kg body wt.) To mitigate the pain threshold and all efforts were made to minimize suffering. The wound was left uncovered during the whole period of the experiment. After 16 days post-wounding the wound tissues were carefully collected under light anesthesia using ketamine (80 mg kg-1, i.p) to evaluate the biochemical, molecular, and histopathological analysis (Keshri, Gupta, Yadav, Sharma, & Singh, 2016).
Experimental design
Even though the majority of wound healing is a physiological process, certain chronic wounds show a significant delay in healing. Often these do not heal perfectly in individuals with low immune profiles. Thus, the present study was undertaken to develop an excision wound model in the immunocompromised state induced by pretreatment with hydrocortisone (HC) 40 mg/kg intramuscularly in male rats. A total of 24 healthy adult Wistar albino male rats weighing 180–200 g were randomly distributed into six groups. Rats were housed under standard conditions (temperature of 25 ±1Ö¯c and relative humidity 50% ± 10%) with 12h light and dark cycle with ad-libitum and standard diet pellets.
- 8- In line 389, write the complete form of "i.m injection". The required has been made (line 408).
- 9- In lines 401 and 404, correct the number "0.5x106cell" and write the power of 6 correctly. The required has been made (line 419 and 422).
- 10- In line 401, "intravenously (i.v)" is written, so in line 404, its full form does not need to be repeated. The required has been made (line 419)
11- How did mice become immunosuppressed? Please explain.
Induction of an immunocompromised condition has been made as follows:
This was accomplished by administering hydrocortisone (40 mg/kg body weight, intramuscularly) daily for one week before wounding, which was continued until the end of the trial to preserve the animals in an immunosuppressed state. HC was dissolved in pyrogen-free sterile water.
- 12- Lines 411-407 are related to statistical analysis and should not be included in this section. In the material and method part of this article, no part is dedicated to statistical analysis. At the end of the material and method, add a section titled "Statistical Analysis" and put these lines in that section. A section of statistical analysis has been made (line 502)
13- In the method section, explain the extract that is finally used on the rats. In line 405, explain more about "formulated extract ointment" and specify the ingredients of this ointment. For example, is the pure extract used on the rat, or is the extract combined with another substance and state the concentration of this extract?
- The alcoholic extract is rupped on the back of the rats. The word ointment was a mistake. The extract applied to the skin is completely pure. (line 417)
14- In the method section, it is said that BMMSCs are taken from rats. Please state this issue in the introduction section so that the reader knows the origin of BMMSCs from the beginning of the article and does not confuse it with human types. The word has been added.
15- The article has writing errors and needs to be revised in this regard. For example, in lines 436 and 437, "10-ml" should be written as "10 ml". In line 439, "70-mm" should be written as "70 mm". Capital letters should not be used in the middle of a sentence. For example, "Flush" in line 438, "Filtered" in line 439, "Cultured" in line 442, "Incubated" in line 443, and similarly in other parts of the article, these items should be corrected and the capital letter changed to small. Thanks Sir., the whole manuscript has been revised. Starting from line 449
- 16- What do you mean by "complete medium" in line 442? Please state the name of the company and the country of this item. In the material and method section, there are many materials and tools used without the name of the company and country, which should be corrected. For example, in line 452, company and country "PBS" should be said. Thanks for the comment. The required is illustrated in section 4.3.1. Additionally, complete medium is an expression known in the world of stem cells which means a complete composition of 1% penicillin, 2% antifungal, bovine serum, etc….) (line 458)
17- BMMSCs culture is mentioned in the method section, but there is no image of the cell culture in the results section. Add images related to cell culture in the results section. The image is added to the supplementary file, under the name of Figure S2 which is mentioned in section 4.3.4. in the manuscript.
18- In line 422 of the method, which is prepared in the explanation of the timing of the ????? ???????, the 3rd, 7th, 10th, and 14th days are given as an example, but in line 135 of the results section, the results at other times (0, 4, 8, 12 and 16 days) has been checked. Please correct the evaluation times of ????? ??????? in the method section. This is corrected in both the results and methods sections.
19- Figure 3 is a graphical evaluation of Figure 2. Merge these two figures together. They are now merged.
In Figure 2, the columns of groups are not well written. What does (-ve) mean in group 1? Group 2 should be written as "induced hydrocortisone (HC)-immunosuppressed". In group 3, "MEBO®" ointment is written, while in the method section of line 395, "povidone iodine ointment" is written, and the name of "MEBO®" ointment is not given. Please write the name of the ointment and the company and country of manufacture in the method section. In groups 4 and 6, only Mesenchymal stem cell is written. Please write the type of stem cells. Also, add "immunosuppressed" in groups 3, 4, 5, and 6. Also in Figure 2, the images of wounds do not have scale bars, so the scale bars of the images should be given.
All the required corrections have been made
- The description of Figure 2 in lines 148-153 should be corrected and it should be specified that rat bone marrow stem cells were used. It is stated now and highlighted. (line 165)
The description of each group for Figure 3 should be corrected as in Figure 2. Figure 3 has no explanation. Please add suitable explanations for these diagrams below the figure. They are merged with figure 2
- 20- Figure 5 shows the histopathology images on which day? In the method section in line 413, it was said that the samples were taken for histopathological evaluation on days 7 and 14, but in the results section, only the results of one day were shown. Please add the pictures of both days 7 and 14.
It has been corrected , in section 4.2.2, as the histology samples are usually taken at the last day of the experiment which is the day 16. (line 427)
Lines 215-231, which are the descriptions of Figure 5, should be given by mentioning the number of each image (1 to 6) in the continuation of line 214 and the description of the figure. They have been corrected. (lines 225-239)
21- The discussion part has started similar to the introduction, while this part should start with the most important and main result of this study and expand, and previous studies related to this result should be reviewed and then other results of this study should be discussed in order of importance. Thanks for your valuable comments. The whole discussion is modified now.
Reviewer 2 Report
It appears that the study “Wound restorative power of Halimeda macrolaba/mesenchy- 2
mal stem cells in immunocompromised rats via Downregulat- 3
ing Inflammatory/Immune cross talk” investigated the potential of combining bone marrow mesenchymal stem cells (BMMSCs) and an algae extract called Halimeda macrolaba to accelerate wound healing in immunocompromised rats. The extract was found to contain chemicals known for their wound-healing properties, while the BMMSCs were characterized for their expression of CD markers. The results showed that the combination of BMMSCs and Halimeda extract significantly improved wound closure, thickness, density of new epidermis and dermis, and skin elasticity compared to the control group. The combination also attenuated oxidative stress, pro-inflammatory cytokines, and NF-KB activation. While the results are promising for regenerative medicine, further safety assessments and clinical trials are needed.
The manuscript has enough nobility. This is a update idea to combine treatments and following their effects to enhance the quality of mediation. Hence the manuscript is new, unique and contemplative.
Comments:
• it would be better if there be more talk about Halimeda macrolaba in introduction.
Author Response
Thanks for your comments Sir. and we are appreciating your nice revision. More data has been added about Halimeda in the introduction section
Reviewer 3 Report
Science seems solid, but it is hard to follow as the structure of the sentences and expressions used are confusing and sometimes ambiguous. The authors use the term “dereplicated compounds”, implying they will use only newly identified secondary metabolites or those with previous history of healing properties. However, in the tests, it seems the whole extract is being used. They seem to be using “dereplicated” as a synonym for “identified”.
The graphical abstract is confusing and crowded. Please make it more succinct.
The introduction does not show a logic flow, going from cytokines to steroids to BMMSCs derived from rats to humans, with no clear sequence.
On 4.2.1, Group 1, the authors state: “intramuscular (i.m) injection of the vehicle of hydrocortisone (HC).” This is very confusing. Why not state the vehicle? (saline?).
The only place where the use of hydrocortisone as the immune-suppressing agent is mentioned is the abstract. No details anywhere else. There seems to be a confusion as to what an “abstract” means.
Please have a fluent English writer/speaker revise this document, with emphasis on the flow of the information presented.
Author Response
Many thanks for the valuable comments

Round 2
Reviewer 1 Report
I have no more comment.
Author Response
Dear Ms. Crystal Chen,
Many thanks for the kind revision of our manuscript (marine drugs-2333700) entitled “Wound restorative power of Halimeda macrolaba/mesenchymal stem cells in immunocompromised rats via Downregulating Inflammatory/Immune cross talk”. Kindly find our highlighted corrected manuscript according to the reviewers’ recommendations. We hope that the new version will satisfy your expectations.
Reviewer 3:
- The authors use the term “dereplicated compounds”, implying they will use only newly identified secondary metabolites or those with a previous history of healing properties. However, in the tests, it seems the whole extract is being used. They seem to be using “dereplicated” as a synonym for “identified”.
Thanks for the valuable comment, the word “dereplicated” was replaced by “identified” and the changes were highlighted.
- The graphical abstract is confusing and crowded. Please make it more succinct.
Many thanks for the valuable comment, the graphical abstract is modified now.
- The introduction does not show a logical flow, going from cytokines to steroids to BMMSCs derived from rats to humans, with no clear sequence.
Many thanks for your nice comment. The introduction’s logical flow was adjusted and highlighted according to the reviewer’s suggestion.
- In 4.2.1, Group 1, the authors state: “intramuscular (i.m) injection of the vehicle of hydrocortisone (HC).” This is very confusing. Why not state the vehicle? (Saline?).
Hydrocortisone injected doses were dissolved in sterile pyrogen-free water, This section has been added to the manuscript and we appreciate this clarification.
The only place where the use of hydrocortisone as the immune-suppressing agent is mentioned is the abstract. No details anywhere else. There seems to be a confusion as to what an “abstract” means.
The section about the Induction of immunocompromised state has been added and its purpose was explained in the manuscript.
- Please have a fluent English writer/speaker revise this document, with emphasis on the flow of the information presented.
Thanks Sir for the valuable comment, the whole manuscript was revised.
Thank you for your consideration of our work,
Sincerely,
Reviewer 3 Report
Small typos (such as "abstract" written twice) still present. The study seems well conducted, but some sentences need to be revised for their construction/meaning, such as the conclusion, where the authors say "The combination of both BMMSCs and Halimeda macrolaba extract and getting the benefit of their immunomodulatory and repairing properties hold a great challenge for modern regenerative medications, especially in complicated cases" I believe the authors meant "promise" instead of "challenge". Some careful proofreading is still necessary before publication.